# Pushing of Magnetic Microdroplet Using Electromagnetic Actuation System

**DOI:** 10.3390/nano10020371

**Published:** 2020-02-20

**Authors:** Georgios Banis, Konstantinos Tyrovolas, Spyridon Angelopoulos, Angelo Ferraro, Evangelos Hristoforou

**Affiliations:** 1Laboratory of Electronic Sensors, National TU of Athens, Zografou Campus, 15780 Athens, Greece; kwstastyr@gmail.com (K.T.); spyrosag@gmail.com (S.A.); ferraro@eie.gr (A.F.); hristoforou@ece.ntua.gr (E.H.); 2Department of Metal Protection and Surface Treatment, National University of Science and Technology (MISiS), 119049 Moscow, Russia

**Keywords:** magnetic driving, magnetic nanoparticles, actuation system, drug targeting

## Abstract

Treatment of certain diseases requires the administration of drugs at specific areas of tissues and/or organs to increase therapy effectiveness and avoid side effects that may harm the rest of the body. Drug targeting is a research field that uses various techniques to administrate therapies at specific areas of the body, including magnetic systems able to drive nano “vehicles”, as well as magnetically labeled molecules, in human body fluids and tissues. Most available actuation systems can only attract magnetic elements in a relatively small workspace, limiting drug target applications to superficial tissues, and leaving no alternative cases where deep targeting is necessary. In this paper, we propose an electromagnetic actuation system able to push and deflect magnetic particles at distance of ~10 cm, enabling the manipulation of magnetic nano- and microparticles, as well as administration of drugs in tissues, which are not eligible for localized drug targeting with state-of-the-art systems. Laboratory experiments and modeling were conducted to prove the effectiveness of the proposed system. By further implementing our device, areas of the human body that previously were impossible to treat with magnetically labeled materials such as drugs, cells, and small molecules can now be accessible using the described system.

## 1. Introduction

The remote manipulation of nano- and microparticles attracts researchers from various scientific fields. Several technologies were adopted, whereby magnetism and composite materials seem to be the most appropriate tools for this task. The applications of magnetic particles in biomedical and clinical research fields are several. They are used for drug delivery in tumors [1,2,3,4], infective diseases [5], thrombosis [6], localized administration of drugs [7], stem-cell therapies [8,9,10], and gene delivery [11,12]. To manipulate magnetic particles, permanent magnets or electromagnets can be used. Permanent magnets propagate high-gradient magnetic fields able to retain magnetic particles in relatively small distances (up to 5 cm); however, the propagated field intensity cannot be adjusted. Therefore, their clinical applications are limited, e.g., for retaining magnetically responsive materials at small distances. A second alternative of magnetic manipulation relies on the use of electromagnets which enable the possibility to fine-tune or switch on/off the generated magnetic field. Many attempts were made to manipulate magnetic agents for clinical purposes such as the Octomag [13] or the Mag-μBot system [14]. Nevertheless, all these systems can only attract magnetically labeled items because of the paramagnetic nature of magnetic nano- and microparticles. Furthermore, an additional obstacle in driving pharmaceutical agents in deep tissues and organs is human body size [15]. In order to reach deep targets while preserving the ability to manipulate pharmaceutical agents, it is necessary to magnetically push nano- and/or microparticles (magnetic injection) deep into the human body. In an early study using permanent magnets with a specific configuration, Sarwar and his colleagues successfully managed to push magnetic particles into rat inner ears, keeping the apparatus at a corresponding “human head” working distance [16,17]. Nevertheless, the system proposed by Sarwar has the major drawback of moving parts. To achieve the pushing effect, all magnetic apparatus must be moved, implying a complex design and a laborious use of the entire device. Electromagnetic driving devices designed and developed by Probst et al. and Bijarchi et al. were also proposed recently [18,19]. However, these electromagnetic devices with adjustable magnetic gradients cannot manipulate magnetic microdroplets at human body working distances.

In this work, we present the design of an actuation system able to manipulate (deflect and push) magnetic particles at relatively big distances from its cores without the need for moving parts. Indeed, the pushing effect is obtained by fine-tuning the electric current feeding a specifically configurated electromagnet array. The optimum design and working conditions (angle, current for each coil, distance, etc.) were optimized by using computer simulations; subsequently, the magnetic system was constructed and tested successfully. This design paves the way for driving pharmaceutical agents in deep areas of the human body. The fact that electromagnets are used offers a promising and fully controllable tool for biomedical applications to push, deflect, or pull magnetic particles at relatively long distances from the magnetic core.

## 2. Materials and Methods

### 2.1. Magnetic System

The distance between the electromagnet and the magnetic particles to be driven is a key parameter in the equation governing any magnetic system that is intended to be used for the remote control of magnetic nano- and/or microparticles [20]. Indeed, the magnetic field intensity must be increased as the distance increases from the magnets. To design our system, we took advantage of the basic principal that, thanks to a specific electromagnet configuration, the magnetic field lines sum together at one specific point due to the linearity of Maxwell’s equations. On this specific point, magnetic lines cancel each other, creating a magnetic field minimum. The magnetic field strength will increase outward from this minimum and create outward forces. The apparatus that was constructed following this reasoning and presented here consists of four coils, two steel cores, four power supplies, and a control system.

To achieve an increasing magnetic field gradient which pushes magnetic particles, the magnetic lines have to cancel themselves (meet at 180°) at a desirable point and, after that, the magnetic field vectors should meet again with an angle different from 180°. We used the finite element method through ANSYS 17 (Ansys, Canonsburg, PA, USA, licensed to National Technical University of Athens) to test different electromagnet arrangements with various angles and various current combinations to optimize the magnetic line alignments and obtain a cancellation node, where only one configuration was selected to be experimentally validated and is presented hereafter. The magnetic core dimensions were 60 × 60 × 170 mm^3^, made from electrical steel SA1008 (Sidenor steel industry, Maroussi, Greece), and the inner and outer coils were wound in 350 and 540 turns, respectively, with 2-mm-diameter copper wire. At each core, two coils were wound (Figure 1). Magnetic cores were surrounded by wooden holders and connected by an adjustable wooden bar to regulate the angle between electromagnetic elements.

### 2.2. Control System

The control system was designed and developed in order to control the characteristics of the supply signal and feed each coil separately with pulsing direct current (DC) (1 s duty cycle, 1 s pulse width). In order to accomplish this, the control system was based on the pulse-width modulation (PWM) technique, according to which the modulation of the width (duration) of a pulsed signal results in the amplitude adjustment of the output voltage. As a result, the designed control system can be used to adjust the current, as well as the pulse width (duty cycle) supplied to each coil. The schematic of the designed circuit is provided in Appendix A (Appendix A). It consists of four channels, each of which can be connected to a different coil. Each channel is controlled by a current mode controller (TS3843B). The output of each controller can be adjusted by a potentiometer, which is located at the upper panel of the control system. A fifth knob is used in order to control the pulse duration, through a selection among 12 different capacitor networks, which are connected to the controller through a Schmitt trigger inverter (74HC14). Finally, the adjusted controller’s output is fed to the gate of a power metal–oxide–semiconductor field-effect transistor (MOSFET) (IRFP3306PbF), which allows the current to flow to the connected coil.

The designed circuit was fabricated on a printed circuit board (PCB), having the required terminals for the connection of the coils. Four power supplies with a maximum output power of 600 W (24 V, 25 A) each were used in order to provide sufficient power to the coils.

### 2.3. Nanoparticles 

For the experiments, a microdroplet of superparamagnetic nanoparticles was used. The product (fluidMAG-Lipid, article number 4119-5) was purchased from Chemicell GmbH, Berlin, Germany. The nanoparticle core was magnetite-functionalized with a matrix phosphatidylcholine, resulting in a hydrodynamic diameter of 100 nm. The surfactant (matrix phosphatidylcholine) is hydrophilic; therefore, nanoparticles were dissolved in water. For the experiment, oleic acid was used in order to force the nanoparticles to form a microdroplet. The nanoparticle buffer was sterile water, and a micropipette with adjustable volume between 0–10 μL was used to transfer the microdroplet of nanoparticles in the oleic acid.

### 2.4. Microscope 

Images and videos were recorded using a simple microscope made of plastic elements to avoid interference with the magnetic field (Bresser junior DM400 digital microscope, Bresser, Germany).

## 3. Results

### 3.1. Magnetic Actuator Modeling and Construction

The electromagnetic system, as well as the control system regulating the current at each coil, was constructed at our laboratory facilities. The validation of the design was done using a Hall sensor in order to compare the generated field with the values from the simulations. Measurements and simulations were taken for 18 A at the outer coils and 6 A in the inner coils. In Figure 2, the whole apparatus is shown together with the simulation results. Experimental measurements are compared and a very small deviation between them can be observed. In Figure 2b, the magnetic field intensity that propagated at a straight line from the middle of the cores is depicted for 18 A at the outer coils and 6 A in the inner coils. At a distance of 7.5 cm from the cores, for the specific values of current in the coils, the magnetic field intensity is minimized due to the opposite direction of the vectors at this point.

### 3.2. Validation of the Pushing Effect

We performed preliminary tests in order to verify the efficiency of the system and the reliability of the simulations. A paramagnetic sphere of 4-mm radius was placed on an aluminum rail at 9 cm, nearby the cancellation node (7.5 cm) but are the point where outward force was maximum (Figure 3a) according to the simulation results as seen in Figure 2. Subsequently, by feeding the coils with 18 A (outer coils) and 6 A (inner coils), the sphere received a magnetic push that moved it by about 10 cm from the initial point (Figure 3b). The results of this experiment confirmed the accuracy of our simulations, as well as the measurements with the Hall sensors. However, since the aim of our study was to demonstrate the possibility to magnetically push small particles in the human body, we set up a second round of experiments.

### 3.3. Remote Control of Magnetic Microdroplet

For this experiment we used a microdroplet with a volume of 10 μL of water-soluble ferrofluid material placed in 100% oleic acid (Figure 4). The petri disc containing the microsphere of ferrofluid nanoparticles and oleic acid was put at a specific distance (9 cm) from the cores in order to align the microdroplet with the point where outward force was maximum (Figure 4), and electromagnets were activated as described previously. Due to the high viscosity of the oleic acid and the size of the specimen, the movement of the microdroplet was restricted only to a few mm as seen in Figure 4. Nevertheless, this significant magnetic push received by the microdroplet demonstrates that it is possible to remotely control paramagnetic material.

Having assured the reliability of the simulations, the next step was to test the deflecting efficiency. To do so, the ferrofluid microdroplet, a cluster of nanoparticles with average size of 100 nm, was put at the same starting point, and the coils were fed with specific currents (Figure 5 and Appendix A, Appendix A). In particular, coil 1 and coil 2 of the left array (Figure 5a,b) were fed with 15.5 A and 4 A, whereas coil 3 and coil 4 of the right array were fed with 4 A and 13 A, respectively. This differential power supplied to the four coils allowed us to create a magnetic flux that moved the droplet toward the *X*-axis (Figure 5c). For the deflecting step that allowed us to move the droplet toward the *Y*-axis (Figure 5d), coil 1 and coil 2 of the left array (Figure 5e) were fed with 17.5 A and 4 A, whereas coil 3 and coil 4 of the right array were fed with 4 A and 11 A, respectively. The differential current suppled to each coil in these experiments allowed remotely controlling a microdroplet of magnetic material; the experimental data were confirmed by the simulations that predicted a change in the direction of the magnetic field vectors (Figure 5b,f) as a consequence of power supplied to the coils. The velocity of the ferrofluid droplet was also calculated to be approximately 135 μm/s. The driving of the magnetic droplet was based on the optical feedback provided by a microscope connected to the computer. However, it is possible to predesign a route and program the actuation system accordingly, in order to drive the microdroplet at a specific route, without intervening during the procedure. The programming of the actuation system will be presented in a future work.

## 4. Discussion

To achieve pushing and deflecting effects of any magnetic material, a controllable cancellation node in a given bidimensional space must be realized. In fact, at this unique point, the magnetic lines at the minimum field point meet each other with opposite vector directions but with the same absolute field intensity values. Since the magnetic fields do not cancel at other points, this point is a unique location of a locally minimum (zero) magnetic field strength. The forces increase from low to high magnetic field; therefore, in the region beyond the cancelation node, the generated forces will push particles away from the magnetic system. An ideal magnetic actuator system should offer the ability to adjust the position of the local minimum point and, therefore, to remotely drive the magnetic objects by regulating the power provided to the electromagnets, and not by moving the magnetic elements.

The magnetic force applied by a non-homogeneous magnetic field on a particle suspended in a fluid is given by the following equation:(1)$$F_{m}=\mu_{ο}\left(\left(m_{p}-m_{f}\right)\cdot \nabla \right)H_{ap},$$
where $\mu_{ο}$ is the permeability of free space, ∇ is the Hamilton operator, $H_{ap}$ is the value of the magnetic field intensity applied at the place where the particle is located, $m_{p}\;\mathrm{is}\;\mathrm{the}\;\mathrm{magnetic}\;\mathrm{moment}\;\mathrm{of}\;\mathrm{the}\;\mathrm{particle},\;\mathrm{and}\;m_{f}$ is the magnetic moment of the liquid where the particle suspends. $\chi_{p}$ and $\chi_{f}$ are the susceptibilities of the particle and the fluid, respectively.
(2)$$m_{p}-m_{f}=V_{p}\frac{3\left(\chi_{p}-\chi_{f}\right)}{\left[\left(\chi_{p}-\chi_{f}\right)+3\left(\chi_{f}+1\right)\right]}H_{ap}.$$
In our case, the $\chi_{f}$ of oleic acid is ~0; thus, the equation can be written as
(3)$$F_{m}=\mu_{ο}V_{p}\frac{3\chi_{p}}{\left(\chi_{p}+3\right)}\left(H_{ap}\cdot \nabla \right)H_{ap}.$$
The force applied by a magnetic field with magnetic field intensity $H_{ap}$ at a magnetic particle of radius α(m) and magnetic susceptibility *χ* is
(4)Fm=4πα33μοχ(1+χ3 )HapdHapdx=2πα33μοχ(1+χ3)∇(|Hap|2).

Based on literature data, the magnetic susceptibility of the magnetic microdroplet is χ ≈ 20 [17] and that of the human body is *χ* ≈ 10^−6^–10^−4^ [21]; thus, the efficiency of the magnetic field on the magnetic microdroplet is not affected remarkably by human tissue. The magnetic force applied depends strongly on the particle size. Very small objects with size on the order of nm can be manipulated by using cooling systems, which allow providing higher currents feeding the coils. Moreover, the magnetic forces can be increased by choosing particles with anisotropic shape [22] or particles with higher magnetic susceptibility. Furthermore, it must be noted that the magnetic force increases according to the increase in the magnetic field gradient. A steeper gradient results in a bigger force on a given microdroplet. Manipulation of magnetic ferrofluids attracted the interest of various scientific fields due to the multidisciplinary nature of applications. Nacev et al. and Torres-Diaz et al. [23,24] described various approaches that were done in order to manipulate ferrofluids; however, they reported the control of ferrofluid at small distances and without the ability to push outward. Other attempts were made by Zhang et al. and Hassan et al. [25,26] to manipulate ferrofluids using uniform magnetic fields which are easier to acquire, but the feasibility of this manipulation applied to human body still presents drawbacks due to system complexity and operational distances. The system presented in this work overcomes many of these issues.

A highly important field of interest is anti-tumor therapy. Indeed, magnetic drug delivery could enable a better treatment efficiency, while minimizing side effects of the chemotherapy. By functionalizing drug nanocarriers with therapeutic molecules able to interact specifically with tumor cells, it is possible to deliver drugs exclusively to the tumor tissue [27]. Another important application is targeted regenerative medicine and, more specifically, magnetically driven stem cells. Stem cells play a special role in renewing and repairing damaged tissues and organs. However, a big obstacle in such therapies is the inability to retain stem cells to the desired location. To achieve this goal, magnetic nanoparticles could be inserted in the stem cells, conferring in this way magnetic properties that allow immobilizing them via an externally applied magnetic field. Furthermore, with magnetically modified cells, it would be possible to avert the problem of the short-term cell engraftment in organs or tissue after the injection. One application of stem cells that is becoming increasingly interesting with significant opportunities in the field of neurosurgery is the treatment of spinal cord injuries. Spinal cord injuries (SCIs) are defined as complicated pathophysiological damages caused by injuries to the spinal column, and there are over 700,000 cases per year worldwide [28]. SCI pathophysiology can be divided into two distinct phases. Primary injury is the initial shearing or compression of the spinal cord tissue. The mechanical force of the primary injury can lead to hemorrhage, disruption of cell membrane integrity, and ion and neurotransmitter imbalance that immediately affects neural function. Secondary injury is related to the progressive inflammatory, ischemic, and apoptotic cascade that follows the initial mechanical assault [29]. The target of stem-cell therapies for SCI is to minimize the spread of secondary injury, improve the function of remaining cell populations, and engage regeneration of neuronal and glial populations. All these therapeutic effects could be achieved via a remote control of magnetically responsive stem cells and an apparatus like that proposed in the present work.

Another interesting field of application of magnetically responsive cells is cellular transplantation, which can be augmented with a combination of growth factors, scaffolds, or other biomaterials that improve cell survival, engraftment, and differentiation. Intraspinal application of these therapies leads to engraftment of transplanted cells, which may promote neural regeneration through several proposed mechanisms [30]. The engraftment on specific locations can be effectively increased by using magnetically modified stem cells. Furthermore, various studies were conducted on the delivery of magnetically modified stem cells at targets such as the myocardium or at the spinal canal, but the targeting in all these studies was primeval, since no study used a sophisticated method to guide and, most importantly, to inject the magnetic stem cells to the location needed [31]. Implementation of our devices could offer a solution for those pathologies that, at the moment, remain without any feasible operative option.

## 5. Conclusions

In this work, we presented an electromagnet system of four coils arranged in two units with a particular configuration. Laboratory experiments and modeling were conducted to prove the effectiveness of the proposed system. We showed, for the first time, an apparatus able to push and deflect magnetic particles just by acting on the currents provided to the four coils without moving any system part. The reason why two units of two coils per core were utilized is essentially due to a better control of the magnetic object during the deflection movement. Indeed, by regulating the power supplied to each coil forming a single unit, the magnetic force generated can be precisely modulated. The pushing effect could be realized with only two coils; however, in order to deflect and pull the microdroplet, the four-coil configuration is necessary. Implementation of our device could offer a solution for pathologies that can be treated with drug target therapies but, at the moment, remain without any feasible operative option.

## Figures and Tables

**Figure 1 nanomaterials-10-00371-f001:**
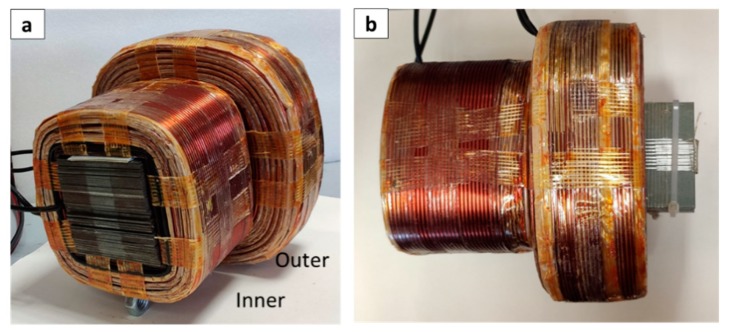
Electromagnet unit arrangement with the two coils (outer and inner) and the electrical steel core: (**a**) front view; (**b**) top view.

**Figure 2 nanomaterials-10-00371-f002:**
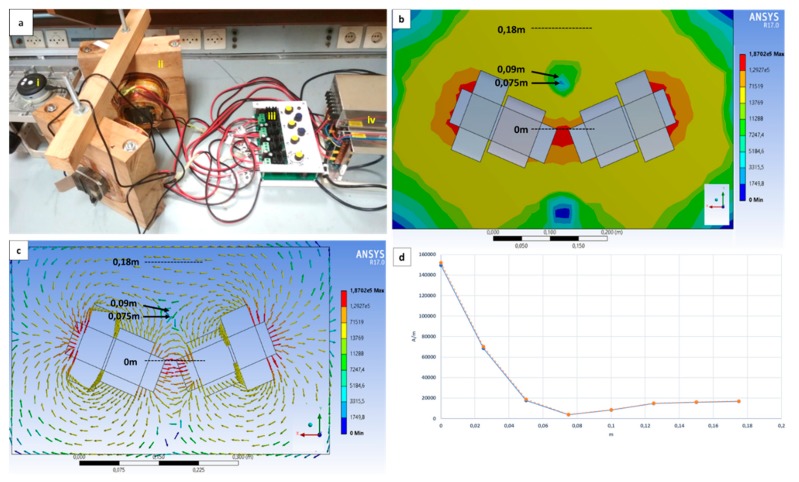
The whole apparatus is depicted in (**a**): (i) the microscope used to monitor the movement of the magnetic droplet; (ii) electromagnets; (iii) the control system; (iv) the power supplies. (**b**,**c**) Simulations for the magnetic field intensity propagated by the electromagnets are shown, as well as the vectors of the field. Arrows in (**b**,**c)** indicate the cancellation area created at given distance from magnet cores and the initial position of the magnetic sphere and microdroplet. (**d**) The magnetic field intensity measured by a Hall sensor (orange color) in comparison with the results of the simulation from ANSYS (blue color). The measurements are taken on the centerline starting from the last point of the cores (dashed lines). The scale bars are in meters.

**Figure 3 nanomaterials-10-00371-f003:**
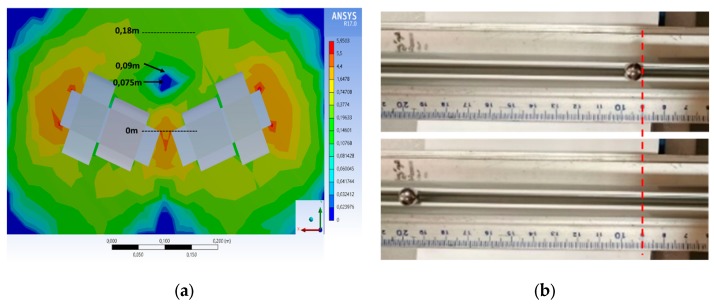
(**a**) Map of magnetic forces generated around the electromagnets; units reported in the simulation are arbitrary and are intended to highlight areas where the force is almost zero (dark blue), while the other colors indicate forces >0. The scale bar is in meters. (**b**) A paramagnetic sphere pushed from 9 to 19 cm on an aluminum rail using the actuator system.

**Figure 4 nanomaterials-10-00371-f004:**
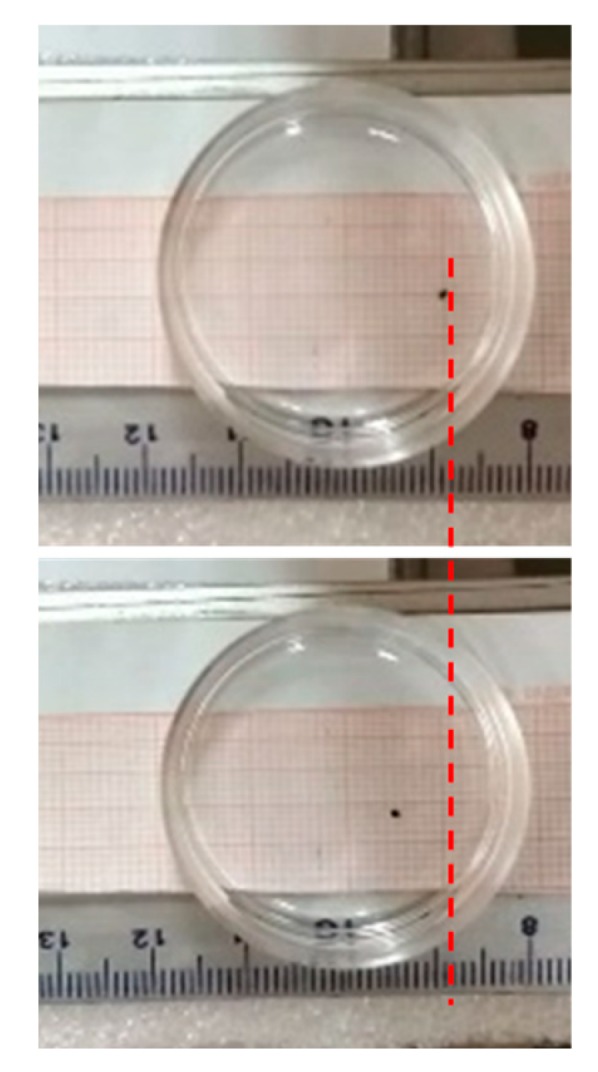
Ferrofluid microdroplet being pushed in oleic acid.

**Figure 5 nanomaterials-10-00371-f005:**
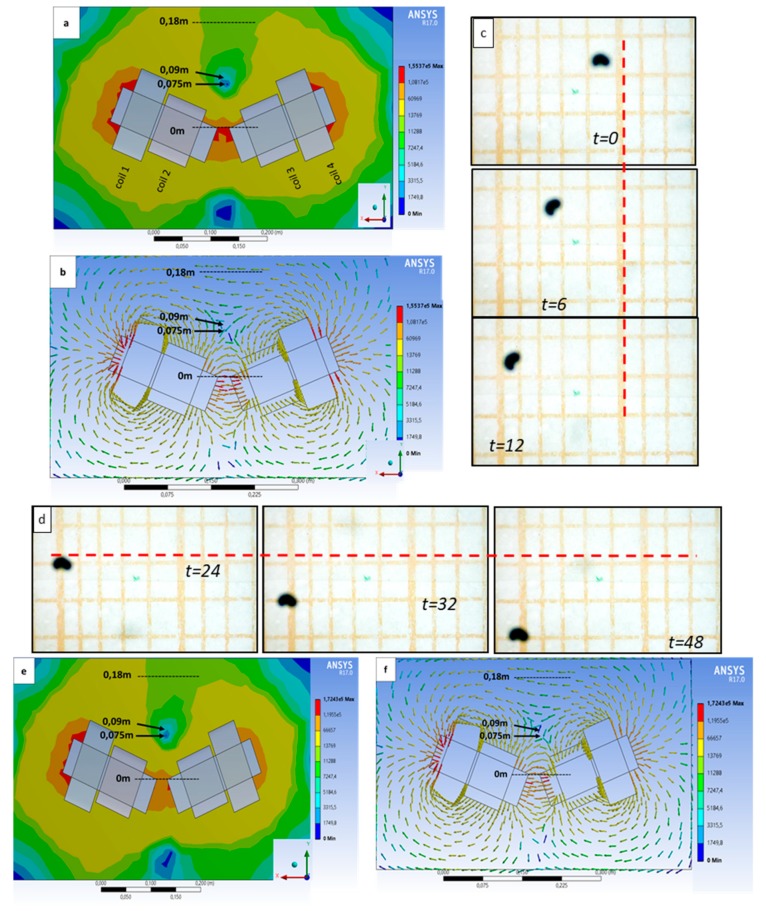
(**a**,**b**) Simulations of the magnetic field intensity and the vectors of the field for currents used to push forward the droplet. (**c**) The positions of the droplet at t = 0, 6, and 12 s. (**d**) The position of the droplet for t = 24, 32, and 48 s. (**e**,**f**) Simulations of the magnetic field intensity and the vectors of the field for currents used to deflect the droplet. Note the change of direction of the vector in the cancellation node (arrows) in (**b**,**f**) due to different current supply to the four coils. The measurements are taken on the centerline starting from the last point of the cores (dashed lines). The scale bars are in meters.

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
