# Peer review of "Pushing of Magnetic Microdroplet Using Electromagnetic Actuation System"

_nanomaterials, 2020, doi:10.3390/nano10020371_

Round 1

Reviewer 1 Report

This paper deals with the development of a novel electromagnetic actuation system for efficient manipulation of magnetic microparticles. The system presents two electromagnets each equipped with two coils whose axes are not aligned but make some angle one with respect to another. The authors pretend to discover an efficient system allowing 2D manipulation of magnetic microparticles (or rather micro droplets) at large distance between the object and the magnets. This is expected to be very beneficial for drug targeting applications. Despite an undoubted practical interest of this work, the paper does give impression of a finished or at least well advanced work.

First, only sub-millimeter objects (as revealed from Fig. 4 and not “clusters of nanoparticles with average size of 100nm” which would not be visible neither by naked eye, nor by an optical microscope because of the optical resolution limit) are manipulated by magnetic fields, requiring electric current supply as high as 6 – 18 A. In drug delivery applications, usually one needs to manipulate much smaller objects, so the question about drug delivery feasibility by the present technique could be risen.

Second, the only set of geometrical parameters of the electromagnets is studied. The effects of the angle between the coils and of the distance between the poles are not reported, such that the studied configuration with the single angle and distance is maybe still not optimal.

The theoretical analysis through eqs. (1) – (6) remains poorly exploited. The authors have to build the maps of the magnetic force distribution and of the particle velocity distributions and compare the latter to experimental results.

Below are listed more specific comments:

Beginning of Sec. 2.1: “the lines should meet again with a nonlinear fashion”. This sentence is unclear and needs to be rephrased. The parent ferrofluid should be better described. What is the liquid carrier for the parent ferrofluid? How the nanoparticles were extracted from the solvent and how were they transferred to oleic acid? 2 and 5: please put a line in (b) and (c) along which the field distribution (d) is reported. Initial position of the droplets should be highlighted on Figs. 2b, c and 5a, b, e, f. The scale bar has to be presented in Fig. 5c, d. What is the role of the inner and outer coils? Why would not just one coil produce a similar field (or field distribution), as compared to two coils by each electromagnet. This point has to be definitely detailed in the manuscript by providing a quantitative comparison of performances of single and double coil arrangements. I suggest displacing the 1st paragraph of the “Discussion” section to Sec. 2.1. “Steer” does not seem an appropriate term for particle displacement. Equations (1) – (5) are classical equations of magnetostatics. Are they so necessary in this paper or maybe a reference to a handbook on electromagnetism could instead be provided? In Eq. (2), the dot product (divH) should be replaced by a cross product (rotH). Homogeneous notations for all vector magnitudes have to be used everywhere in the paper: either bold non-italic symbols or arrows above non bold italic symbols. Please choose one of these two styles in accordance with “Instructions for Authors” of the Journal. How was the value of the ferrofluid microdroplet susceptibility (about 20) assessed? Usually, reliable values of the magnetic susceptibility are difficult to measure. In general, it is better to speak everywhere about magnetic microdroplets instead of magnetic microparticles. The last two paragraphs of the Discussion section are much more appropriate for the Introduction Section. The Supplementary Information file referring to “Supplementary Figure 1” seems to be missing.

In view of the above comments, I do not recommend the paper for publication in the present form. The authors are advised to thoroughly revise the above mentioned points before future submissions.

Author Response

Reviewer 1

This paper deals with the development of a novel electromagnetic actuation system for efficient manipulation of magnetic microparticles. The system presents two electromagnets each equipped with two coils whose axes are not aligned but make some angle one with respect to another. The authors pretend to discover an efficient system allowing 2D manipulation of magnetic microparticles (or rather micro droplets) at large distance between the object and the magnets. This is expected to be very beneficial for drug targeting applications. Despite an undoubted practical interest of this work, the paper does give impression of a finished or at least well-advanced work.

Comment. First, only sub-millimeter objects (as revealed from Fig. 4 and not “clusters of nanoparticles with average size of 100nm” which would not be visible neither by naked eye, nor by an optical microscope because of the optical resolution limit) are manipulated by magnetic fields, requiring electric current supply as high as 6 – 18 A. In drug delivery applications, usually one needs to manipulate much smaller objects, so the question about drug delivery feasibility by the present technique could be risen.

Answer. The reviewer is right; however, this is just a proof of concept that must be further developed. One solution to work with smaller samples could be to use a cooling system for the coils which can allow even higher current supply. In this way, small objects in the order of nanometers can be manipulated. Another solution would be to drive objects with higher magnetic susceptibility (higher susceptibility particles or even microrobots (ref 14) which may allow drug delivery to deep targets. Finally, shape anisotropy (see ref 22) also may help. We modified the discussion part by adding a sentence and a ref. explaining alternative solutions to overcome the feasibility of small objects’ manipulation. The following sentence has been added: “Smaller objects can be manipulated by using cooling systems for the coils and thus, higher currents feeding the coils. Also, the magnetic forces can be increased by choosing particles with anisotropic shape [22] or particles with higher magnetic susceptibility.”

Comment. Second, the only set of geometrical parameters of the electromagnets is studied. The effects of the angle between the coils and of the distance between the poles are not reported, such that the studied configuration with the single angle and distance is maybe still not optimal.

Answer. Even though only one configuration is presented, several have been studied. Indeed, we optimized the system through ANSYS software to study the effect of several angles between the coils and distance between poles. Therefore, the presented one is the best to have cancelation node at a distance of 7,5 cm with adequate magnetic field gradient after that point. In Material and methods section, we added a sentence explaining that many configurations have been studied and the optimum has been chosen for experimental validation. The following integration has been added: “…..different electromagnets arrangements with various angles and various current combinations to optimize the magnetic lines alignments and obtain a cancellation node, only one configuration was selected to be experimentally validated and is present hereafter”.

Comment. The theoretical analysis through eqs. (1) – (6) remains poorly exploited. The authors have to build the maps of the magnetic force distribution and of the particle velocity distributions and compare the latter to experimental results.

Answer. Thank you for the comment. We improved the theoretical analysis by providing the equations needed to define the magnetic force. We can provide a qualitative magnetic force distribution map (attached file only for reviewers) but a quantitative would not have meaning because in the magnetic force equation there should be taken into account the size and the susceptibility of the microdroplet, something that could not apply in the simulation.

On the other hand, the velocity distribution map cannot be asserted. An acceleration map could be extrapolated though, qualitatively, which would be again similar to the magnetic field intensity and also similar to the distribution map of the magnetic force.

Below are listed more specific comments:

Comment. Beginning of Sec. 2.1: “the lines should meet again with a nonlinear fashion”. This sentence is unclear and needs to be rephrased.

Answer. Thank you for the comment. The sentence has been rephrased as following: “…the lines magnetic field vectors should meet again at an angle different than 180°”

Comment. The parent ferrofluid should be better described. What is the liquid carrier for the parent ferrofluid? How the nanoparticles were extracted from the solvent and how were they transferred to oleic acid?

Answer. The paragraph 2.3 has been modified by adding details about nanoparticles buffer (sterile water) and the way used to transfer them in the oleic acid (micropipette).

Comment. 2 and 5: please put a line in (b) and (c) along which the field distribution (d) is reported. Initial position of the droplets should be highlighted on Figs. 2b, c and 5a, b, e, f. The scale bar has to be presented in Fig. 5c, d.

Answer. Fig 2 and fig 5 have been modified by marking the points between the reference distance (0m) and the initial position of the droplet (0,9m) together with all sampled points up to 0,18m, according to figure 2d.

Comment. What is the role of the inner and outer coils? Why would not just one coil produce a similar field (or field distribution), as compared to two coils by each electromagnet. This point has to be definitely detailed in the manuscript by providing a quantitative comparison of performances of single and double coil arrangements.

Answer. Thank you for the comment. The reason for use 4 coils is for a better steering performance. We provide the simulation made for 2 coils instead of 4 (attached file only for reviewers). As seen, the cancelation node still exists so the microdroplet can be pushed as well. However, the magnetic field gradient is lower than previously but most importantly, without the inner cores, it is not possible to regulate the field propagated at all the sides of the cores thus, the microdroplet cannot be pulled or deflected without the other 2 coils. Summing up, in order to pull the magnet, the inner coils are needed, in order to deflect the microdroplet, the outer coils must be fed with different currents and in order to push the microdroplet, 2 coils would be enough.

Comment. I suggest displacing the 1st paragraph of the “Discussion” section to Sec. 2.1.

Answer. Thank you for the advice. We moved the paragraph as suggested.

Comment. “Steer” does not seem an appropriate term for particle displacement.

Answer. Thank you for the advice. We analyzed several terms and we concluded that deflection is the most appropriate to describe the movement we caused on the microdroplet (besides the pushing).

Comment. Equations (1) – (5) are classical equations of magnetostatics. Are they so necessary in this paper or maybe a reference to a handbook on electromagnetism could instead be provided? In Eq. (2), the dot product (divH) should be replaced by a cross product (rotH). Homogeneous notations for all vector magnitudes have to be used everywhere in the paper: either bold non-italic symbols or arrows above non bold italic symbols. Please choose one of these two styles in accordance with “Instructions for Authors” of the Journal.

Answer. As suggested the equations 1-5 have been removed, all the vector magnitudes have been homogenized and eq 6 is being explained better.

Comment. How was the value of the ferrofluid microdroplet susceptibility (about 20) assessed? Usually, reliable values of the magnetic susceptibility are difficult to measure.

Answer. The susceptibility of the particles is similar to that used by Shapiro et al. (ref 17).

Comment. In general, it is better to speak everywhere about magnetic microdroplets instead of magnetic microparticles.

Answer. Thank you for the advice. We replaced microparticles with microdroplets everywhere in the text as well as in title’s paper.

Comment. The last two paragraphs of the Discussion section are much more appropriate for the Introduction Section.

Answer. We appreciate the suggestion but by moving also this part, the manuscript will be imbalanced, very long introduction and poor discuss. We hope the reviewed will understand our point.

Comment. The Supplementary Information file referring to “Supplementary Figure 1” seems to be missing.

Answer. We will upload it, sorry for the inconvenience.

In view of the above comments, I do not recommend the paper for publication in the present form. The authors are advised to thoroughly revise the above-mentioned points before future submissions.

Reviewer 2 Report

The manuscript submitted by Georgios Banis et al. reports on the development of double electromagnet-based system to actuate nanoparticles. It presents a substantial advantage with respect to the work of Shapiro et al. (ref.17) by application of electromagnets, which allows the control of ferrofluid droplet without mechanical motion of the system. On the other hand, with respect to the more recent uncited papers by Probst et al. (10.1016/j.jmmm.2010.08.024) and Bijarchi et al. (10.1016/j.sna.2019.111753), the system discussed has the advantage of large magnet-particle distance, which is suitable for manipulation of (super)paramagnetic objects at the distance of few centimetres. As such it is pretty well suited to in-vivo applications. However, the manuscript is difficult to follow. It is mainly due to extensive grammar and spelling errors as well as low quality of the figures and discussion. Therefore, the manuscript could be considered for publication only upon significant improvements, which are listed in the following:

1. Since the topic is not broad, the discussion should not omit similar works of other authors. These include the two paper mentioned earlier as well as the seminal review papers by Nacev et al. (10.1109/MCS.2012.2189052) and Torres-Diaz and Rinaldi (10.1039/c4sm01308e).

2. It is difficult to realize the differences between data shown in Fig.2b, Fig.4a, Fig.4e and Fig.2c, Fig.4b, Fig.4f, respectively. Thus, addition of the line profiles, equivalent to Fig.2d, in Fig.4 is suggested. In addition, the scale (and units) in all the panels of Fig.2 and Fig.4 is unclear - this must be included. The exact position (direction), at which the dependence shown in Fig.2d was probed, should be marked in Fig.2b or Fig.2c.

3. The formulas 1-5 should rather be omitted - can be found in any textbook (reference can be provided). The reference for the derivation of the formula (6) should be given instead. Force exerted by particles, as derived using formula 6 and distribution Fig.2b could be included.

4. The discussion is rather empty. It would significantly improve if the dependence of the current (or magnetic field) applied to each electromagnet during the actuation shown in Fig.4 (and movie) is plotted against the coordinates. It could be quantitatively compared to the results obtained using permanent magnets (10.1016/j.jcis.2014.01.044). How much it differs from the expected values discussed by Shapiro et al. (ref. 17)? Discussion would greatly improve as well, if the results obtained are compared, at least qualitatively, to the behaviour of iron oxide ferrofluid droplets in sheer flow under uniform field (10.1039/c8sm02522c and 10.1063/1.5047223).

5. Detailed proofreading and language correction is necessary. There are numerous grammar errors and typos present. Just one example: in Fig.2 captions "dash cycles" should read "dashed circles".

Author Response

Reviewer 2

The manuscript submitted by Georgios Banis et al. reports on the development of double electromagnet-based system to actuate nanoparticles. It presents a substantial advantage with respect to the work of Shapiro et al. (ref.17) by application of electromagnets, which allows the control of ferrofluid droplet without mechanical motion of the system. On the other hand, with respect to the more recent uncited papers by Probst et al. (10.1016/j.jmmm.2010.08.024) and Bijarchi et al. (10.1016/j.sna.2019.111753), the system discussed has the advantage of large magnet-particle distance, which is suitable for manipulation of (super)paramagnetic objects at the distance of few centimetres. As such it is pretty well suited to in-vivo applications. However, the manuscript is difficult to follow. It is mainly due to extensive grammar and spelling errors as well as low quality of the figures and discussion. Therefore, the manuscript could be considered for publication only upon significant improvements, which are listed in the following:

Comment 1. Since the topic is not broad, the discussion should not omit similar works of other authors. These include the two paper mentioned earlier as well as the seminal review papers by Nacev et al. (10.1109/MCS.2012.2189052) and Torres-Diaz and Rinaldi (10.1039/c4sm01308e).

Answer 1. Thank you for the advice. We included the suggested works in the new version by mentioning them in the discussion.

Comment 2. It is difficult to realize the differences between data shown in Fig.2b, Fig.4a, Fig.4e and Fig.2c, Fig.4b, Fig.4f, respectively. Thus, addition of the line profiles, equivalent to Fig.2d, in Fig.4 is suggested. In addition, the scale (and units) in all the panels of Fig.2 and Fig.4 is unclear - this must be included. The exact position (direction), at which the dependence shown in Fig.2d was probed, should be marked in Fig.2b or Fig.2c.

Answer 2. We modified figure 2 and 5 in order to make them clear. Additional information are now available in each panel of the figures as well as in legends.

Comment 3. The formulas 1-5 should rather be omitted - can be found in any textbook (reference can be provided). The reference for the derivation of the formula (6) should be given instead. Force exerted by particles, as derived using formula 6 and distribution Fig.2b could be included.

Answer 3. Thank you for the advice. The formulas were replaced by the ones explaining the equation (6 (now 4))

Comment 4. The discussion is rather empty. It would significantly improve if the dependence of the current (or magnetic field) applied to each electromagnet during the actuation shown in Fig.4 (and movie) is plotted against the coordinates. It could be quantitatively compared to the results obtained using permanent magnets (10.1016/j.jcis.2014.01.044). How much it differs from the expected values discussed by Shapiro et al. (ref. 17)? Discussion would greatly improve as well, if the results obtained are compared, at least qualitatively, to the behaviour of iron oxide ferrofluid droplets in sheer flow under uniform field (10.1039/c8sm02522c and 10.1063/1.5047223).

Answer 4. We are grateful to the reviewer for the advices. We thought to build the discussion around possible biomedical applications of our system, the topic is in line with the special issue aim. Definitively a comparison with other systems it would be interesting, however we are planning to publish a broader analysis along with the control system where the comparison will be included.

Comment 5. Detailed proofreading and language correction is necessary. There are numerous grammar errors and typos present. Just one example: in Fig.2 captions "dash cycles" should read "dashed circles".

Answer 5. The new version has been reviewed and edited by an expert of scientific language. We believe now that English and presentation are fluid and correct.

Round 2

Reviewer 1 Report

The authors have responded to most or the Reviewer questions. A few issues still remain to be adressed in the paper before final acceptance for publication:

  1. The force map presented in Fig. 2 of the response letter seems to be important for this study and I suggest including it to the manuscript (even in aritrary units as long as the magnetic susceptibility of the drop is not known). The spatial resolution of the force map must be significantly improved.
  2. The conclusion section must be provided with a short but complete summury of the main results of the work.
  3. mp and mf in Eq. (1) should be highlighted in bold as vectors.
  4. Supplemental Materials contaning "Supplemental Figure 1" cited in Sec. 2.2  seem still not be uploaded. The only Supplemental Materials file contains a marked-up manuscript not destined for publication. Plesase upload an appropriate file for Supplemental Materials!

Author Response

We thank once again the reviewer for taking time to go through the revised version. We have replied to all comments and we think that now the third version has been improved a lot thanks to their comments and advices. In the following text we put our answers and explain all changes with respect to the second version.

The authors have responded to most or the Reviewer questions. A few issues still remain to be adressed in the paper before final acceptance for publication:

Comment 1. The force map presented in Fig. 2 of the response letter seems to be important for this study and I suggest including it to the manuscript (even in aritrary units as long as the magnetic susceptibility of the drop is not known). The spatial resolution of the force map must be significantly improved.

Answer 1. Thanks for the suggestion. We improved the resolution of the map and included it in the main text in figure 3a. A legend has also been added to figure 3.

Comment 2. The conclusion section must be provided with a short but complete summury of the main results of the work.

Answer 2. The third version of our manuscript now includes a dedicated paragraph with the conclusion.

Comment 3. mp and mf in Eq. (1) should be highlighted in bold as vectors.

Answer 3. Thanks for noticing it, we now corrected mp and mf as vectors

Comment 4. Supplemental Materials contaning "Supplemental Figure 1" cited in Sec. 2.2  seem still not be uploaded. The only Supplemental Materials file contains a marked-up manuscript not destined for publication. Plesase upload an appropriate file for Supplemental Materials!

Answer 4. We cross checked and now the supp figure 1 is uploaded online.